Chicks change their pecking behaviour towards stationary and mobile food sources over the first 12 weeks of life: improvement and discontinuities

Murphy Kenneth J. 1 2 3 ken.murphy@kcl.ac.uk
Hayden Thomas J. 1
Kent John P. 2
1 School of Biology and Environmental Science, University College Dublin , Belfield, Dublin , Ireland
2 Ballyrichard House, Arklow , Co. Wicklow , Ireland
Vallortigara Giorgio
3 Current affiliation: Department of Forensic and Neurodevelopmental Science, The Institute of Psychiatry, Psychology and Neuroscience, Kings College London, London, United Kingdom

Electronic publication date: 2014 Oct 23
Publication date: 2014
Volume: 2
Electronic Location ID: e626
Received 2014 Jul 26; Accepted 2014 Sep 27
Copyright: © 2014 Murphy et al.
Copyright year: 2014
Copyright holder: Murphy et al.
License: This is an open access article distributed under the terms of the Creative Commons Attribution License, which permits unrestricted use, distribution, reproduction and adaptation in any medium and for any purpose provided that it is properly attributed. For attribution, the original author(s), title, publication source (PeerJ) and either DOI or URL of the article must be cited.
License URL: https://creativecommons.org/licenses/by/4.0/

Keywords: Chicks, Nutritional requirements, Pecking, Peck rate, Development

Funding: The author declares there was no funding for this work.

==============================
Chicks (Gallus gallus domesticus) learn to peck soon after hatching and then peck in rapid bursts or bouts with intervals of non-pecking activity. The food sources may be static such as seeds and chick crumb, or mobile such as a mealworm. Here, changes with age in pecking toward chick crumb and a mealworm were measured.

Chicks were reared in pairs and their pecking of crumb food was video recorded in their pair housed environment, from food presentation, every third day from day 8 (wk 2) to day 65 (wk 10). Peck rate at crumb food reached maximum levels at day 32 (wk 5), and then declined, fitting a quadratic model, with no sex, sex of cagemate, or box order effects. Within bouts the peck rate was higher and it increased to day 41 (wk 6) and then declined, and here males pecked faster than females. A change in dietary protein concentration from 22% to 18% at day 28 (wk 4) had no effect on subsequent peck rate.

Pecking at and consumption of a mealworm in pair housed chicks were measured weekly from wks [5 to 12]. The latency to first worm peck and latency to swallow decreased to wk 8 and increased thereafter. The peck rate to first wormpeck and number of pecks to swallow increased to wk 8 and then declined paralleling the changes with crumb food. The increase in peck rate is coupled with an increase in efficiency in worm catching.

The results are consistent with the view that the improvement in pecking ability and accuracy compliments change in nutritional requirement best served by an invertebrate food (IF) source requiring speed to achieve feeding success, especially with live prey. When this food source is no longer crucial these associated skill levels decline. An appreciation of the role of domestic fowl in controlling insect populations, at farm level, that are often vectors in disease spread is lacking.

Introduction

All five of the jungle fowl species are classified as highly omnivorous, eating a wide variety of food items of plant and animal origin throughout the year with seeds often only a minor component of the diet (Klasing, 2005). Food of animal origin that is frequently consumed include termites, their eggs and pupae, winged ants, grasshoppers, spiders, moths, beetles and grubs (Collias & Collias, 1967; Savory, 1989; Klasing, 2005) with the breeding season in one study of wild jungle fowl matching the annual cycle of termite availability (Collias & Collias, 1967). The nutritional needs of chicks change with age, requiring a higher protein diet, when young, for growth and development, and lower levels for maintenance during adulthood, and this is reflected in the changing dietary habits (Klasing, 1998). Gallinaceous and especially Gallus gallus young consume high levels of invertebrate food (IF) during the first two months of life and levels decline thereafter, regardless of insect availability (Savory, 1974; Savory, 1980; Savory, 1989; Savory, Wood-Gush & Duncan, 1978). However, it should be pointed out following Savory (1989) that while the proportions of IF decline with age in the growing chick the quantities consumed can continue to increase especially within the first two months of life, and the increasing quantities will require more time spent eating. This is important as insects provide protein and vitamin B12, essential for growth that are almost non-existent in plant proteins (Klasing, 1998).

Domestic fowl adjust their nutritional intake depending on their physiological requirements (Hughes, 1984). Broilers can self-select the appropriate combination from different protein mixtures to maximise growth (Gous & Swatson, 2000) and can adapt to diet changes within 24 h at all ages (Yo et al., 1998). Thus, chicks of different species can adjust their food intake appropriate to nutrient requirements (Hale & Green, 1988; Covasa & Forbes, 1995; Shariatmadari & Forbes, 1993; Forbes & Shariatmadari, 1994; Yo et al., 1998; Klasing, 1998). In Sage Grouse (Centrocercus urophasianus) an increase in insects in the diet leads to increased survival and growth (Johnson & Boyce, 1990).

In the feeding context recent studies have shown the considerable cognitive abilities of the chick reflecting their phylogenetic and ecological history. For example, Rugani, Vallortigara & Regolin (2014) show that chicks are able to distinguish between numerical comparisons using training and test stimuli that consisted of static 2D images with a number of black squares one of which was usually associated with a hidden food reward. Chicks also have a leftward bias when locating a food source after training to peck at either the fourth or sixth position in a series of 16 identical aligned positions, when the training apparatus was rotated by 90 degrees (Rugani et al., 2010).

Chicks are pre-disposed to peck at small three-dimensional objects (Dawkins, 1968; Hogan, 1973). Pecking incorporates precisely coordinated, visual, neuromuscular and tactile processes, that has been studied intensively in domestic chicks (Yo et al., 1998; Yo et al., 1997; Picard et al., 1997) and hens (Hutchinson & Taylor, 1962). Pecking involves the isthmo-optic visual system, which is used to focus on moving prey and is better developed in ground feeding birds (Miles, 1972). The relative myopia of the frontal fields adapts in the growing chick as the distance from ground to beak changes (Hodos & Erichson, 1990). During pecking the chick’s head remains in a static position for 75% of the time and two out of three pecks at wheat grain and pellets did not result in feed intake (Yo et al., 1997), and thus a distinction between eating and pecking is made and these pecks are assumed to be exploratory, and not included in their calculation of ‘eating rate’. Here, chicks are housed in pairs as feeding especially in younger birds takes place in a social context involving the mother hen in the first few days of life and thereafter siblings and/or conspecifics (for review see Nicol, 2004).

At the behavioural level, discontinuities in development are recognised especially in play behaviour (Bateson, 1981; Bateson & Martin, 2013) and in social organisation of domestic fowl (Kent et al., 2009) and behaviours appropriate to a particular stage of development in young chicks with the broody hen, can and do fall out of the behaviour repertoire when no longer functional (Workman & Andrew, 1989; Vallortigara et al., 1997). While much is known about satisfying the dietary need of chicks in an agricultural production context little is known as to how these needs are facilitated at a behavioural level. The concern here is with developmental changes in the pecking behaviour of the chick with a static and a mobile food sources. Different food sources may require different eating strategies.

We measured changes in pecking behaviour over the first 12 weeks of life in Brown Leghorn type domestic chicks. Crumb food was used to study peck rate at static stimuli and mealworms were used as the mobile food source. We hypothesised that rapid ballistic type pecking should develop to facilitate capturing mobile IF that best meets the nutritional needs of the growing chick.

This research complies with the current laws of this country, and the ethical guidelines for the use of animals in research as outlined by The Association for the Study of Animal Behaviour.

Method

Study 1—Changes in peck rate toward a static food source (chick crumb) from day 8 to day 65

The peck rate during the first minute from food crumb presentation from day 8 (wk 2) to day 65 (wk 10) and changes in the within bout peck rate in pair housed chicks were measured using video recording and associated technology.

Subjects and procedure

Thirty-two (13 male and 19 female) incubator hatched domestic chicks (Brown Leghorn X) from 4 batches with eight chicks per batch were randomly assigned to and reared in pairs in wooden boxes (49.3 × 46.2 × 35.6 cm high) with a wire mesh front and roof. The chicks in this study were obtained from a semi feral population of brown leghorn X hens with recent jungle fowl ancestry and are thus genetically close to the original jungle fowl (see Rubin et al., 2010) and were maintained in a flock of approx. 45 females with a fluctuating number of males from which the eggs were obtained to produce chicks for this study. One pair (1 male and 1 female) was removed due to developmental delay in one chick. Chicks were colour leg banded for identification. The floor was covered with brown paper and straw and water was provided ad libitum. A crumb food diet (22% protein to day 28, then 18% protein crumb diet—in accord with meal provider recommendations for growing chicks) was provided in a 15.5 cm diameter dish. A window provided natural illumination and in addition electric light (60 watt) was provided until 10 pm each evening. In addition heat was provided with a 150 watt infrared electric light bulb over each box as required, providing additional light.

For testing, food dishes were removed at 10 pm, boxes cleaned and water replaced. Some, but minimal food would have remained in the box environment. Next morning, from 10 am, chicks were videorecorded (UC3000 8 mm video camcorder) in their pair house environment, in a random order, from food presentation, for twenty minutes. This procedure was repeated every third day from day 8 to 65. Videotapes were converted to media files (i.e., CDs) using Broadway software, and analysed using the Observer Video Pro v4.0 behaviour analysis system (Noldus Inc., 1993).

Peck rate during the first minute (± 5.12 s) of feeding

The peck rate during the first minute was measured by subtracting the latency to first peck (L) from the duration (Mean = 61.28 s, Range = 60.24–65.32, S.D. = 0.853) to give the time spent pecking (T). The number of pecks (P) was divided by the time (sec) spent pecking (T−L) to give a peck rate per second and then multiplied by 60, and expressed as pecks per minute (p/min). Pecks/Min=P/Tsec−L×60.

Peck rate within bouts of pecking

To eliminate the effect of time spent in non-pecking activity and give a more accurate measure of pecking capacity a within-bout peck rate was measured. The first bout or series of 6–8 consecutive pecks occurring 30 s after the first peck was identified. The time to first peck of this series (T1) was subtracted from time to the last peck (T2), i.e., the duration of the bout. No inter-peck interval within the bout was more than half the inter-bout interval on either side of the selected bout (see Machlis, 1977). The number of pecks (P) was divided by duration (sec) of the bout (T2−T1), multiplied by 60 to give peck rate per minute and was measured for days 11 (wk 2), 29 (wk 5), 41 (wk 6), 53 (wk 8) and 65 (wk 10). Bout peck rate/min.=P/T2−T1sec×60.

Statistics

Repeated measures ANOVA in SAS v8.0 examined changes in peck rate over time, and within-subjects contrasts and post hoc pairwise comparisons were calculated in SPSS v8.0 to determine effects of batch, sex, sex of cagemate, and position of rearing box. Latency to peck and latency to swallow were log transformed to normality, whereas number of pecks did not transform to normal. Repeated measures ANOVA examined changes in the pecking behaviours.

Results

Peck rate during first minute

The data is normal (Kolmogorov–Smirnov Z = 0.733; p = 0.656). Peck rate changed significantly over the 10 weeks in all batches (F = 20.25; df = 1.557; p < 0.001), rising gradually to day 32/wk 5; see Fig. 1 and Table 1). No sex (p = 0.248), sex of cage mate (p = 0.818), or box order effects were found (p = 0.612). Batches 2 and 4 were hatched earlier in the season (mean day 171.5 days—mid June) than batch 1 and 3 (mean day 262—mid September) and peck rate peaked earlier (2 and 4, mean 30.5 days/wk 5; 1 and 3, mean 42.5 days/wk 7; F = 15.59; df = 3.23; p < 0.001).

Figure 1 Peck rate (per min.); n = 30 chicks during the first minute after food presentation every third day from day 8 (wk 2) to day 65 (wk 10).

Table 1 Peck rate (per min) for the 1st minute after food presentation; n = 30 chicks from day 8 (wk 2) to day 65 (wk 10).

Age (days)	Age (wks)	Peck rate (pecks/min.)	S.D.	Range	N	
8	2	70.86	17.05	60.38	27	
11	2	73.98	19.19	70.75	30	
14	2	88.24	18.47	76.27	28	
17	3	88.62	22.86	83.33	28	
20	3	99.57	21.86	86.04	30	
23	4	94.67	21.36	90.10	30	
26	4	106.44	28.62	101.85	29	
29	5	111.45	26.81	112.24	30	
32	5	121.63	24.06	95.77	30	
35	5	112.46	29.76	131.65	30	
38	6	119.0	30.74	124.67	30	
41	6	119.42	21.13	106.18	30	
44	7	115.58	23.70	104.28	30	
47	7	99.32	32.03	132.61	30	
50	8	107.52	21.21	85.44	30	
53	8	105.57	25.24	94.9	30	
56	8	95.21	20.29	75.35	30	
59	9	92.47	26.98	115.62	30	
62	9	94.31	21.23	68.22	30	
65	10	93.53	29.41	139.13	30	

Bout peck rate

Within bout peck rate increased to day 41 (wk 6) and then declined. The weeks differ significantly (F = 4.12; df = 1.113; p < 0.05) and the trend was also quadratic like that of the minute rate (F = 9.097; df = 1, 16; p < 0.01; r2 = 0.7628), with no batch (p = 0.211), cagemate (p = 0.899), or box order (p = 0.697) effects, though males had a higher peck rate than females (F = 4.52; df = 1.23; p = 0.044) (see Table 2 and Fig. 2).

Figure 2 Within bout peck rate (per min.); n = 30 chicks on five occasions from day 11 (wk 2) to 65 (wk 10).

Table 2 Mean within bout, peck rate after 30 s of food crumb pecking, n = 30 chicks.

Age (days)	11	29	41	53	65	
(wks)	2	5	6	8	10	
Peck rate (pecks/min)	151.16	170.01	201.37	178.99	166.30	
S.D.	46.25	43.99	38.58	45.76	30.78	
N	30	30	30	30	30	

Discussion

Peck rate changes are significantly over the 10 weeks of the study for both total duration of pecking which increased to day 32 (wk 5), and then declined and within bout pecking which increased to day 41 (wk 6) and then declined. It is the decline in peck rate that is the most interesting and worthy of explanation and is considered below. Earlier hatched chicks (June) had a peck rate peak at an earlier age and this is consistent with more rapid development rate in chicks hatched early in the year and is attributed to the long days experienced early in the year in the northern hemisphere accelerating development (Morris & Fox, 1958). However both June and September chicks show the same developmental trends. Within bout pecking was higher in males and this may compensate for greater time spent in vigilance behaviour in male chicks (Murphy, 2005).

Study 2—The effect of a change in dietary protein concentration on the peck rate of chick crumb

As the diet used in study 1 was changed from a 22% protein crumb diet to an 18% protein crumb diet on day 28 (wk 4) it was necessary to control for the possible effects of this diet change on subsequent behaviour. Thus the following experiment was carried out.

Subjects and procedure

24 chicks from 2 batches of 12 (n = 10 males; 14 females) were pairhoused as described in study 1 from day 8 (wk 2) to 38 (wk 6). The diet of 6 pairs, three pairs from each of the two batches (Gp A) were maintained on a 22% crumb to day 38 while the diet of Gp B was changed to an 18% protein crumb diet on day 28 (wk 4) as in experiment 1. Peck rate was measured on the day before the change (day 27) on the day after the change (day 29) and then at three-day intervals to day 38.

Results

Difference in peck rate between the two groups over the five test occasions (F = 0.407, df = 4.88, p = 0.803), and between sexes (F = 0.063, df = 1.20, p = 0.805) were not significant. A decline in peck rate on the day following food change in the experimental Gp B, was found. There were no significant difference between the two groups on the day before (day 27, Group A mean = 100.9, SD = 26.4; Group B mean = 108.3, SD = 33.9; t = −0.599, df = 22, p = 0.555) or on the day after the food protein change (day 29, Mean group A = 103.55, SD = 103.6; Group B = mean = 97.1, sd = 22.0, t = 0.621, df = 22, p = 0.541).

Discussion

While changing the protein concentration of food from 22 to 18% protein was followed by a slight drop in peck rate (Group B) this was not significant and there was no differences in the peck rate between the two groups either before or after food change. Thus the decline in peck found in study 1 cannot be attributed to change in diet protein.

Study 3—Changes in peck behaviour towards a mobile food source, the mealworm (Tenebrio molitor)

Here the concern is with changes in pecking behaviour, including latency to peck toward a living and moving invertebrate—the mealworm—as the chick ages to wk 12. The mealworm is similar to if not the same as what chicks would encounter in natural settings.

Subjects

10 incubator hatched chicks (4 hatched on 10 September (2m and 2f), and 6 hatched on 17 September (4f and 2 m)) were group-reared in their age groups to weeks five and four respectively. They were colour leg banded and transferred randomly in pairs to the rearing boxes described above, though in a large open shed. They could not see other chicks in the adjacent boxes during rearing or testing. Natural lighting was provided and electrical light was made available until 10 pm. They were fed ad libitum 22% crumb to day 28, and from then 18% crumb diet. Food dishes were removed at 10 pm and on the following morning testing took place from 10 am in a random order.

Procedure

Chicks were each fed five mealworms on two consecutive evenings prior to each test. For testing chicks were separated by a wire screen in the centre of their rearing boxes on the morning of the test. Testing took place from week [5 to 12] at weekly (7 day) intervals. Two mealworms were dropped from above, on to the floor of boxes simultaneously, one for each chick by an assistant seated behind the rearing boxes. When the mealworms were eaten and not less than 15 s after first presentation the procedure was repeated. If a chick did not eat the mealworm the procedure was repeated 1 min after the previous presentation. The procedure was repeated three times on each test day (i.e., three mealworms for each chick). Chicks were videorecorded and behaviour measured from the presentation of the first mealworm. Videotapes are converted to media files and analysed using the Observer 4.0 as described above. The following were measured; latency from the mealworm landing onto the floor to the first wormpeck, latency to swallowing, number of pecks to swallowing and peck rate to first wormpeck.

Results

Latency to peck, number of pecks to swallow and latency to swallow were not normally distributed and were log transformed. Number of pecks did not transform to normal. Repeated ANOVA and Friedman’s nonparametric test with post hoc comparisons were used to examine changes in the pecking behaviours toward the mealworm. One chick was removed from the study as it swallowed only one mealworm during the first seven tests.

Latency to first peck

The latency to first peck differed significantly between weeks decreasing to wk 8 and then increasing (see Table 3), reflecting increased efficiency to wk 8 for catching live prey (F = 3.208, df = 7.35, p = 0.01). Wk 5 differed from wk 8 (p = 0.013) and wk 9 (p = 0.041), and the overall trend was quadratic (F = 79.146, df = 1.5, p < 0.001). There were no sex differences (F = 0.096, df = 1.52, p = 0.766).

Table 3 Pecking at a live mealworm; latency to first peck, latency to swallow, number of pecks to swallow and peck rate; n = 10 chicks.

Age (weeks)	5	6	7	8	9	10	11	12	
Latency to first peck									
Mean (sec)	2.87	2.73	1.77	1.14	1.37	1.71	1.25	1.55	
S.D.	2.73	3.61	2.04	.83	1.49	1.79	0.75	1.57	
Latency to swallow									
Mean (sec)	3.22	3.08	2.21	1.48	1.58	2.01	1.44	1.80	
S.D.	2.92	3.75	2.57	.88	1.58	1.8	0.98	1.69	
No. pecks to swallow									
Mean	1.19	1.29	1.35	1.45	1.25	1.32	1.17	1.3	
S.D.	.63	.66	.69	.96	.72	.48	.38	.47	
Peck rate									
Mean (pecks/min)	65.03	63.36	77.39	84.49	80.11	69.03	70.76	80.32	
S.D.	74.11	55.37	57.62	58.38	46.99	44.47	48.37	52.93	
N	9	9	9	8	8	9	9	7	

Latency to swallow

Latency to swallow changed between weeks decreasing to wk 8 and then increasing, reflecting greater efficiency in catching live prey to wk 8 (F = 2.707, df = 7.35, p = 0.024). Wk 5 took longer than wk 8 (p < 0.05) and wk 9 (p = 0.028) and the overall trend was quadratic (F = 41.222, df = 1.5: p = 0.001; see Table 3). There were no significant sex differences (F = 0.307, df = 1.52, p = 0.597).

Number of pecks to swallow

The number of pecks to swallow did not differ between weeks (χ2 = 12.027, df = 7, p = 0.1; see Table 3) and this may be due to the small number of pecks required to catch the mealworm, though males took significantly fewer pecks to swallow (F = 14.443, df = 1.52, p = 0.007).

Discussion

As chicks aged they became more efficient with a decrease in latency to peck and to swallow the live prey to wk 8 and from wk 8 these latencies increased. Males took significantly fewer pecks to swallow and this is consistent with a significantly greater within bout pecking rate at crumb food (study 1) in males.

General Discussion

Chicks learn to peck soon after hatching and then peck in rapid bursts or bouts with intervals of non-pecking activity (Machlis, 1977). Young chicks need a higher protein diet for growth and this is reflected in dietary habits (Klasing, 1998; Klasing, 2005). We show that the chick’s pecking rate for both static and mobile food sources develops with age and reaches maximum peck rate levels at wk 5 for crumb food, with the within bout peak rate reaching maximum levels at wk 6. For mealworms the efficiency in terms of reduced latency to first peck and the reduced latency to swallow a mealworm, improved to wk 8 and was coupled with an increase in the number of pecks to swallow and an increase in the peck rate to wk 8. After wk 8 there was a discontinuity of these efficiencies. These increases in peck rate to crumb food and to a mealworm show that the chicks are getting more skilled and better prepared to capture the mobile IF with time.

The diet of young gallinaceous birds in the first two weeks of life and in Gallus gallus young in the first two months of life contain a high proportion of IF (Savory, Wood-Gush & Duncan, 1978; Savory, 1989; Klasing, 2005). From wks [1 to 4] between 50 and 90% of the diet in young Gallus gallus chicks in naturalistic environments consists of IF and from wks [5–8] their diet was still between 10% and 50% IF (see Savory, Wood-Gush & Duncan, 1978; Savory, 1989). It should be noted that as the birds grow, similar proportions would mean greater quantities, and would require increasing efficiencies to maintain the same or even a declining proportion of IF consumption.

Chicks can select the required dietary needs in a choice situation when offered different meal diets with differing protein levels (Hughes, 1984; Kaufman, Collier & Squibb, 1978; Gous & Swatson, 2000; Hale & Green, 1988). In the natural situation IF is part of a variety of food types consumed by this highly omnivorous species (Collias & Collias, 1967; Savory, 1989; Klasing, 2005). High protein IF are usually available during the reproduction season in most avian species (Klasing, 1998; Savory, Wood-Gush & Duncan, 1978). In Northwest India, the main flights of termites are in June/July, when young junglefowl chicks are plentiful (Collias & Collias, 1967). In a feral population the adult fowl diet from January to July consists mainly of grasses and seeds and from August to December, mainly oats. By July, the diet of the chicks hatched in May consisted of over 50% IF. In autumn insects were less plentiful, and the diet of the growing feral chick was very similar to that of the adult, though significantly, younger chicks continued to consume higher levels of IF (Savory, Wood-Gush & Duncan, 1978).

It is important to note that the results of study 1 show that chicks hatched earlier in June did show a peck rate during the first minute of food presentation earlier at wk 5 while September hatched chicks showed a peak peck rate at wk 7. However both groups showed the same pattern of improvement to peak followed by a decline. The mealworm tested chicks that were also hatched in September show their peak pecking levels at wk 8 followed by the discontinuity. Seasonal factors have long been known to influence levels of growth and development in fowl (Morris & Fox, 1958) and more recently artificial light has been shown to advance the development of seasonal avian reproductive physiology in an urban context in blackbirds (Dominoni, Quetting & Partecke, 2013). Further, in imprinting studies the importance of drawing a distinction between developmental age (age since incubation commenced) and chronological age (age since hatching) on performance has long been appreciated (Gottlieb, 1961; Gottlieb, 1963), and in a study of maternal behaviour the necessity of drawing the same distinction when comparing twin born calves (shorter gestation) with single born calves (longer gestation) has been noted (Kent, 1987). Season of birth does have an effect on the course of development as seen here in study 1 with the early hatch chicks showing a peak peck rate earlier. However, the developmental pattern with an increase in skill levels up to a certain age and then followed by a decline, or rather a discontinuity, is robust across the seasons. It should be pointed out that sharply timed behavioural changes in chicks and broody hens over the first five weeks since hatching have been reported for other behaviours such as ground scratching and dustbathing (Workman & Andrew, 1989). They also showed that chicks run ahead of the hen with particular behaviours increasing at a certain age, and then declining sharply so that an explanation in terms of progressive maturation as part of development is unlikely (Workman & Andrew, 1989; Vallortigara et al., 1997). So the decline in peck rate found here, after a period of progressive increase should not be seen as counter intuitive but rather as part of normal development in the chicks.

The findings here complement what is known about the changing nutritional needs and associated behavioural habits of growing fowl in the natural and conventional environments. The changing dietary needs for a high proportion of IF in growing chicks is facilitated at the behavioural level by increased pecking efficiencies enabling the young chick to meet their then nutritional needs. When a high IF diet is no longer necessary, the skill levels previously required decline.

Males had a higher within bout peck-rate when eating crumb than females. Males also took fewer pecks to swallow a mealworm. Males engaged in more frequent and longer periods of vigilance behaviour in this feeding context (Murphy, 2005). The role of vigilance during feeding behaviour by male chicks may be best understood in a wider developmental and ecological context where males–male and male–female inter-individual-distance increases especially during the endogenous, testosterone driven juvenile phase wks 16–26 (Kent et al., 2009) and the role of vigilance in this changing social context deserves further elaboration.

Further, in the omnivorous domestic fowl an appreciation of their role in controlling insect populations, at farm level is lacking. Such insect populations are often vectors in the spread of disease (see Godfray, 2013) and fowl in the farmyard and in tropical and in sustainable bio-diverse contexts (see Broom, Galindo & Murgueitio, 2013) may play an important and unappreciated role in their control through their eating behaviour and dietary habits.

Supplemental Information

Data S1 Peck rate data

Click here for additional data file.

Table S1 Mean within bout, peck rate after 30 s of food crumb pecking, n = 30 chicks

Click here for additional data file.

We would like to thank Dr. Finian Bannon for statistical assistance, Dr. Domhnall Jennings for advice on using The Observer 4.0 and Mary Ann Kent for assistance with graphics.

Additional Information and Declarations

Competing Interests

Author Contributions

Animal Ethics

The authors declare there are no competing financial or other interests. John Kent is a researcher/owner at Ballyrichard House.

Kenneth J. Murphy performed the experiments, analyzed the data, contributed reagents/materials/analysis tools, wrote the paper, prepared figures and/or tables.

Thomas J. Hayden contributed reagents/materials/analysis tools, reviewed drafts of the paper.

John P. Kent conceived and designed the experiments, analyzed the data, contributed reagents/materials/analysis tools, reviewed drafts of the paper.

The following information was supplied relating to ethical approvals (i.e., approving body and any reference numbers):

This research complies with the current laws of this country, and the ethical guidelines for the use of animals in research as outlined by The Association for the Study of Animal Behaviour.

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
