# Peer review of "Chicks change their pecking behaviour towards stationary and mobile food sources over the first 12 weeks of life: improvement and discontinuities"

_PeerJ, doi:10.7717/peerj.626_

## Round 0.1 · original submission · Minor Revisions

Both Reviewers liked your paper but recommended some minor revisions. Please pay attention in particular to providing the non-specialist readers with adequate background.

Reviewer 1 ·

Basic reporting

No comments

Experimental design

It is not clear why Authors decided to use these formula to calculate the peck rate or the peck feeding. Do the Authors can add some References or describe better their choices?

Validity of the findings

No comments

Additional comments

I have really appreciated the topic discussed in this article, but I strongly recommend the Authors to describe better the behavioral and the cognitive characteristics of the animal model that they have employed in their study. In this way the article would be more interesting to a wider number of readers, that maybe do not know well this bird.
For example, it is well known that day-old domestic chicks prefer to peck symmetric rather than non-symmetric objects (see Elena Clara et al.).
Day-old domestic chicks are also able to remember up to 3 minutes where a food item (e.g. a mealworm) has disappeared (Regolin et al 2005). Day old chicks can also master numerical discrimination: for example they prefer 4 mealworms over 1 (Rugani et al. 2013), showing that they can discriminate between quantities. I also think that should be interesting to discuss the use, in scientific literature, of the mealworms to train day-old chicks to solve some cognitive problems. Using mealworms as reinforcement, chicks can learn to identify the container, in a series of identical containers, that contains the food reinforcement. They can learn to identify the third, the forth or the six container in a series of 10 identical (Rugani et al. 2007, 2010, 2011).

In the end a curiosity: The Authors found a difference in the peck rate between the chicks hatched earlier in June with respect to those hatched in September. Did the Authors find any difference between these two groups of chicks? For example did they notice differences of weigh, or something else?

Reviewer 2 ·

Basic reporting

The ms is clearly framed despite poor in terms of broader background. For instance, Authors may be interested in citing works investigating chicks' foraging habits and their development depending on social interactions which, in natural environments, are of primary importance (e.g., studies reviewed in Nicol 2004, Learning & Behavior, 32(1): 72-81).

- is the word "compliment" correct (lines 33, 367)? According to the definition of the oxford dictionaries, should be complEment, instead
- please check "et al.," in the main text, which is not always followed by "dot" and "comma"
- line 357, please add a full stop after the word seasons
- please check journal abbreviations / format in the reference list

Experimental design

Why did Authors checked for the effect of the position of cages? Were they kept in different places (i.e., some closer to sources of smell, heat, light)? The position is not specified in the description of the methods.
Why the foraging activity of the first week was not analyzed - if considered that chicks belong to precocial species (line 302)? Similarly, why the tenebrio molitor was inserted only starting from the 5th week onward?
Why in Exp 3 Authors tested ten birds only reducing to 1/3 the sample size from Exp 1?

Validity of the findings

It is not clear whether eggs were supplied from a commercial hatchery or rather small local farms: this may have implications on the effects discussed. The observed difference between batches may depend on differences between roosters' propensity to forage rather than months of hatching.
Effects of domestication (e.g., Rubin et al., 2010, Nature, 464: 587-591) should be discussed in the first case at least, since commercial rearing has overcome a series of *limitations* of natural seasonal breeding.

---

## Round 0.2 · accepted · Accept

I believe you successfully answered all issues raised by the Reviewers and that the paper can be published.